# Molecular and Cellular Mechanisms Underlying the Cardiac Hypertrophic and Pro-Remodelling Effects of Leptin

**DOI:** 10.3390/ijms25021137

**Published:** 2024-01-17

**Authors:** Morris Karmazyn, Xiaohong Tracey Gan

**Affiliations:** Department of Pharmacology and Physiology, University of Western Ontario, London, ON N6A 5C1, Canada

**Keywords:** leptin, cardiac hypertrophy and remodelling, heart failure, intracellular signalling, autophagy, mitochondrial dynamics

## Abstract

Since its initial discovery in 1994, the adipokine leptin has received extensive interest as an important satiety factor and regulator of energy expenditure. Although produced primarily by white adipocytes, leptin can be synthesized by numerous tissues including those comprising the cardiovascular system. Cardiovascular function can thus be affected by locally produced leptin via an autocrine or paracrine manner but also by circulating leptin. Leptin exerts its effects by binding to and activating specific receptors, termed ObRs or LepRs, belonging to the Class I cytokine family of receptors of which six isoforms have been identified. Although all ObRs have identical intracellular domains, they differ substantially in length in terms of their extracellular domains, which determine their ability to activate cell signalling pathways. The most important of these receptors in terms of biological effects of leptin is the so-called long form (ObRb), which possesses the complete intracellular domain linked to full cell signalling processes. The heart has been shown to express ObRb as well as to produce leptin. Leptin exerts numerous cardiac effects including the development of hypertrophy likely through a number of cell signaling processes as well as mitochondrial dynamics, thus demonstrating substantial complex underlying mechanisms. Here, we discuss mechanisms that potentially mediate leptin-induced cardiac pathological hypertrophy, which may contribute to the development of heart failure.

## 1. Introduction

An adipocyte-derived satiety factor was identified by the Jeffrey Friedman research group using obese (*ob*/*ob*) mice in which the obesity is caused by a mutation in the leptin gene, resulting in complete leptin deficiency. This finding in 1994 [1] heralded a new era of research now generally referred to as adipobiology. This factor, named leptin (from the Greek word *leptos*, meaning thin), is a 16 kDa protein synthesized primarily, but not exclusively, by white adipocytes and possesses a principal function to supress appetite by its direct actions on the hypothalamus [2,3]. Thus, leptin was initially considered as a potential effective treatment for obesity via its satiety effects. Clinical trials aimed at reducing obesity with leptin administration have proven to be disappointing overall, however, likely owing to the fact that obese individuals demonstrate elevated circulating leptin levels (hyperleptinemia) due to increased adiposity and the presence of leptin resistance in these individuals [3]. Indeed, plasma leptin concentrations have been shown to be closely correlated to the degree of adiposity [4]. Since the discovery of leptin, a large number of other adipocyte-derived proteins have been identified, generally referred to as adipokines, possessing a myriad of biological effects [5,6]. These are summarized in Table 1, although prominent among these is adiponectin, which, in a general sense, exerts effects opposite to those seen with leptin [5,7]. Although leptin unquestionably plays important roles in a number of cardiac pathologies such as myocardial ischemia/infarction, in this review, we will restrict our discussion to leptin, the most studied member of the adipokine family, and its effects on the heart, which are likely of particular importance to our understanding of pathological cardiac hypertrophy and remodelling per se.

Cardiac hypertrophy is generally considered as an adaptive process in response to various forms of myocardial stressors such as ischemia/infarction, pressure overload, as well as chemical-induced stress by hormones, pathogens, and many other factors [8]. The cardiac hypertrophic response to these stressors is referred to as pathological hypertrophy, resulting in myocardial remodelling associated with cardiomyocyte hypertrophy and cardiac fibrosis that can eventually evolve into heart failure [8,9]. Pathological cardiomyocyte hypertrophy represents the result of a complex series of events including alterations in signal transduction mechanisms as well as upregulation of pro-hypertrophic genes in response to diverse stimuli [8]. While some overlap between the two types of hypertrophy exists, in general, pathological hypertrophy is distinct from physiological hypertrophy, which, as the term implies, is the result of physiological stimuli such as chronic exercise, does not include extracellular remodelling such as fibrosis, and can be associated with improved cardiac function [9,10]. The possible contribution of leptin to pathological hypertrophy is the main focus of this review.

## 2. Leptin Receptors and Underlying Cell Signalling Processes: General Perspectives

From a general perspective, leptin exerts its biological effects through binding to the leptin receptor or ObR (also referred to as LepR although the ObR designation will be used throughout this text as a matter of consistency), a Type I cytokine receptor of which there are six isoforms produced by alternative splicing [11,12,13]. As illustrated in Figure 1, all ObRs have identical extracellular domains but differ substantially in terms of intracellular domains, which account for cell signalling activation. Thus, the most important of these receptors in terms of biological effects of leptin is the long form (ObRb), which possesses the complete intracellular domain of 302 amino acids, which is linked to full cell signalling processes. In contrast, the function of the short form of the receptor (ObRa) is likely to facilitate the transport of leptin across the blood–brain barrier although this is not known with certainty. Other short forms of ObR (ObRc, ObRd, ObRf) possess intracellular domains of various lengths ranging from 32 to 40 amino acids, thus exerting limited biological responses. A soluble ObR (ObRe) has also been identified, which lacks an intracellular domain, is not anchored to the cell membrane, and is secreted into the circulation. ObRe likely participates in leptin transport and thus can regulate plasma leptin concentrations [14]. ObRs are expressed ubiquitously in a variety of tissues, which accounts for the multifaceted nature of the protein. The architecture of ObRs has recently been reported and suggests a complex two-step mechanism for ObR activation in which leptin first binds to the receptor via the high affinity site cytokine-binding homology region 2, thus forming a 1:1 complex followed by interaction between leptin and the ObR Ig domain, which results in the formation of the site 3 interface, thereby dimerizing to form a 2:2 complex, resulting in cell signalling activation [15].

From a general perspective, binding of leptin to ObRb, and to a lesser degree some short forms of the ObR, results in the activation of various intracellular signalling processes, producing a myriad of biological responses in a variety of tissues including the heart, as discussed below. Among the most recognized of these responses is the activation of the Janus kinase-signal transducer and activator of transcription (JAK/STAT) pathway, which is well established as an important mediator of the biological effects of numerous cytokines and growth factors [16]. Four JAK isoforms have been identified whereas seven STAT isoforms exist, which act as transcriptional regulators. Thus, JAK/STAT activation and the resultant effects on gene expression may be important determinants of cardiac pathologies, particularly those involving hypertrophic responses such as myocardial remodelling and heart failure (see below). A summary of leptin-dependent JAK/STAT activation is illustrated in Figure 2.

Other leptin-activated cell signalling processes have also been identified in various tissues and cell types [13]. Among these are various protein kinases including phosphoinositide 3-kinase (PI3K), protein kinase B and C (PKB, PKC), extracellular-signal-regulated kinase (ERK), mitogen-activated protein kinases (MAPKs), as well as the p115Rho guanine nucleotide exchange factor-RhoA/Rho-associated, coiled-coil-containing protein kinase-dependent mitogen-activated protein kinase (RhoA/ROCK) pathway. Those pathways, which have been shown to mediate cardiac actions of leptin, will be discussed in greater detail below. 

## 3. ObR Antagonists

The elucidation of molecular structures of ObRs and mechanisms of ObR activation has led to the development of relatively specific receptor antagonists, which have aided in the development of our understanding of the role of leptin in physiological and pathophysiological processes. Various forms of ObR antagonists have been developed including leptin mutants as well as short-chain peptides, which consist of parts of the original leptin sequence. In general, both the leptin mutants as well as short-chain peptides can bind to ObR but do not have the ability to activate the receptor. Pegylated forms of these antagonists have also been developed, which improve in vivo pharmacokinetic properties, resulting in increased efficacy compared to non-pegylated antagonists. In addition, antibodies to ObR have also been developed but these possess some limitations particularly as potential therapeutic agents due to poor in vivo absorption following administration. Thus, substantial progress has been achieved in developing effective ObR antagonists with extensive ongoing research in this field. For more detailed discussion of the development of ObR antagonists, and modulators in general, various reviews can be recommended for interested readers [17,18,19].

## 4. Leptin and Cardiovascular Disease in General

Leptin likely participates in the development of cardiovascular diseases through both indirect and direct processes. With respect to the former, the metabolic effects of leptin are well established and include proinflammatory and atherogenic effects [20]. Although hyperleptinemia is a close indicator of increased adiposity and obesity, which are risk factors for the development of cardiovascular disease, there still exists some uncertainty as to a cause-and-effect relationship between leptin and the development of cardiovascular pathologies. This includes the potential direct effect of leptin on cardiac pathology, as discussed below. Nonetheless, leptin has been proposed as a potential link between obesity and the development of cardiovascular disease by modulating a number of components of the cardiovascular system such as the heart and vasculature [21,22]. With respect to the latter, a number of studies have shown that hyperleptinemia is associated with endothelial dysfunction, particularly under conditions of obesity [23,24]. ObRb- and leptin-induced endothelial dysfunction have also been identified in the coronary arteries of various species particularly at high concentrations of the protein synonymous with hyperleptinemia [25]. Despite the ability of leptin to produce endothelial dysfunction, this study failed to demonstrate a vasoconstricting effect of the peptide [25]. Of particular relevance to this review, a number of clinical studies have shown a close relationship between hyperleptinemia and the magnitude of left ventricular hypertrophy, as well as being discussed in Section 5.1. 

## 5. Leptin and Cardiac Function and Dysfunction

The direct effects of leptin on cardiac function can be categorized into two specific areas. The first involves acute effects of leptin possibly representing physiological responses but which can also contribute to cardiac pathology, as noted below. The second represents chronic responses to leptin exposure resulting in pathological changes such as cardiomyocyte hypertrophy, thus contributing to the development of cardiac pathology including heart failure. The ability of leptin to directly affect cardiac performance is supported in theory by the fact that both leptin and its receptors are present in cardiac tissue [26] as well as the fact that 60 min exposure to leptin can directly depress contractile function in isolated mouse ventricular myocytes through a mechanism involving the endothelin-1 receptor and NADPH oxidase activation [27]. Moreover, as will be discussed in greater detail in Section 6, many of the cell signalling components noted above and others that are linked to ObR activation are expressed in cardiac cells. 

Leptin has been shown to acutely modulate energy metabolism in the heart, particularly that related to glucose and fatty acid oxidation. For example, 60 min treatment with leptin at a concentration of 60 ng/mL increased fatty acid oxidation in isolated working rat hearts by approximately 60% although glucose oxidation rates were unaffected [28]. Importantly, the increased fatty acid oxidation rates were not associated with increased cardiac work although oxygen consumption increased by 30% whereas cardiac efficiency was decreased by 42%. The underlying mechanism was found to be unrelated to AMP activated protein kinase (AMPK) activation as that seen in skeletal muscle, highlighting the fact that the metabolic effects of leptin may be governed to a large degree by tissue specificity as well as other factors [29]. Using a similar isolated working heart model, other investigators demonstrated that leptin-induced increased fatty acid oxidation was dependent on STAT3-nitric oxide-p38 MAPK activation and was also associated with depressed cardiac function [30]. Studies using an HL-1 cardiomyocyte line similarly demonstrated increased fatty acid oxidation and uptake although the former effect only occurred with short-term, i.e., one-hour, exposure to leptin whereas fatty acid uptake by these cells was maintained for the 24 h maximum treatment period [31]. In contrast to increased cardiac fatty acid oxidation induced by direct effects of leptin, glucose oxidation was found to be unaffected following leptin administration in isolated working rat hearts [28]. The stimulation in cardiac fatty acid oxidation by leptin was suggested to be dependent on enhanced fatty acid translocase (FAT/CD36)-mediated fatty acid uptake, resulting in increased fatty acid oxidation [32]. Taken together, these findings suggest that leptin may contribute to reduce cardiac lipotoxicity via its ability to stimulate fatty acid oxidation. Indeed, enhanced cardiac accumulation in *db*/*db* ObR-deficient mice was prevented in these mice in which cardiac ObR was specifically re-expressed in cardiomyocytes [33]. This result is similar to that seen in a mouse model of lipotoxic cardiomyopathy produced by cardiomyocyte-specific overexpression of the acyl CoA synthase gene in which hyperleptinemic conditions reduced the degree of lipotoxicity [34]. Overall, it appears that leptin may be an important regulator of cardiac energy metabolism particularly through its ability to regulate fatty acid oxidation and thus potentially functions as an endogenous lipotoxicity inhibitory factor.

### 5.1. Leptin and Cardiac Hypertrophy

Evidence from both clinical and experimental studies strongly suggests a link between the development of left ventricular hypertrophy and leptin. Thus, a number of clinical studies reported a close positive relationship between left ventricular hypertrophy and plasma leptin levels [35,36], although such studies do not prove a cause-and-effect relationship. Moreover, high plasma leptin concentrations have been used as predictors of increased severity of heart disease including heart failure [37]; however, other reports showed an inverse relationship between plasma leptin levels and left ventricular mass [38,39,40].

More convincing results supporting a role of leptin come from experimental studies. In this regard, the majority of studies have shown a hypertrophic or cardiomyocyte hyperplasia response following leptin administration, thus demonstrating a direct hypertrophic effect of leptin [41,42,43,44,45,46,47,48,49]. Induction of myocardial overexpression during ischemia and reperfusion has been shown to enhance myocardial remodelling, resulting in increased myocardial dysfunction [50]. Moreover, leptin has been shown to contribute to myocardial remodelling by stimulating myocardial fibrosis, thus further contributing to the development of heart failure [50,51]. 

Interestingly, leptin has also been proposed to function as an autocrine factor via its ability to mediate the pro-hypertrophic effect of both angiotensin II and endothelin-1 [52]. Supporting this concept, at least with respect to angiotensin II, a recent study showed that the antihypertrophic effect of the angiotensin receptor blocker telmisartan in hypertensive rats was due to inhibition of the autocrine function of leptin [53]. As leptin is produced by adipocytes as well as by cardiac tissue, it therefore serves as both an autocrine and paracrine modulator of cardiac function, contributing to the complexity of its underlying actions. Indeed, as recently reviewed, autocrine regulation of cardiac remodelling reflects the contribution of a plethora of cellular processes [54]. Thus, as summarized in Figure 3, leptin-induced cardiac changes can reflect locally derived leptin synthesized within the heart in either a paracrine or autocrine process or circulating adipocyte-derived leptin. 

### 5.2. Proposed Cardiac Beneficial Effects of Leptin

Although the major focus of the present review involves the underlying mechanisms contributing to the pro-hypertrophic and pro-remodelling effects of leptin, it should be noted that various reports suggest a beneficial effect of the protein in terms of reducing myocardial hypertrophy and remodelling. These studies generally involved genetic mouse models expressing either leptin (*ob*/*ob* mouse) or leptin receptor (*db*/*db* mouse) deletions. Indeed, studies using these animal models as well as animals with cardiac-specific ObR deletions have generally demonstrated enhanced remodelling and left ventricular dysfunction, thus suggesting a beneficial effect of endogenous leptin [55,56,57]. The reasons for such discrepancies are uncertain but may involve multiple factors including an experimental model, chronic versus acute experimentation, animal species, as well as other factors. It is possible that an intact leptin system may serve as a cardioprotective mechanism, which would account for the deleterious responses seen in *ob*/*ob* or *db*/*db* mice. However, excessive leptin production such as that seen in severe obesity conditions would contribute to the cardiac hypertrophic and remodelling processes. Clearly, further work is required to delineate the precise effect and role of leptin in myocardial hypertrophy in various experimental models including those in which leptin or leptin signalling is modified through genetic manipulation.

## 6. Intracellular Signalling Pathways Underlying the Pro-Hypertrophic Effects of Leptin

Based on a survey of existing literature, there is substantial evidence that the cardiac hypertrophic effects of leptin are mediated through a multiplicity of potential cellular mechanisms involving alterations in cell signalling, induction of autophagy, as well as changes in mitochondrial dynamics, as illustrated in Figure 4. This diversity of effects may reflect the mechanistic nature underlying hypertrophy produced by leptin or it may be determined using an experimental model, animal species, or leptin concentration and dose used in a particular study.

### 6.1. MAPK Activation

The first demonstration of a hypertrophic effect of leptin originated from the authors’ laboratory and showed that 50 ng/mL of leptin induced a potent hypertrophic response when added to cultured neonatal rat ventricular myocytes [41]. We initially focused on the potential role of the MAPK system in view of extensive evidence from various laboratories showing that MAPK, particularly p38 MAPK, activation represents a major cell signalling pathway for induction of cardiac hypertrophy [58,59,60] and it has been proposed that p38 MAPK inhibition represents a potentially effective approach towards mitigating heart failure [61]. We showed that both phospho-p38 and phospho-p44/p42 MAPKs were rapidly, although transiently, increased following leptin addition with maximum effects seen 5 and 10 min after leptin addition. Although both phospho-p38 and phospho-p44/p42 MAPKs were elevated, only the p38 MAPK inhibitor SB203580 completely abrogated the hypertrophic response whereas the p44/42 MAPK inhibitor PD98059 was without an effect [41]. This finding is in agreement with our report of leptin-induced RhoA/ROCK resulting in p38 nuclear translocation as discussed in Section 6.3. 

### 6.2. Endothelin/ROS Pathway Upregulation

Another potential pathway underlying the hypertrophic effects of leptin and using a similar experimental model as described above stem from a study by Xu et al. [42] who proposed that the hypertrophic effect of leptin was due to upregulation of reactive oxygen species (ROS) and ET-1 levels in myocytes exposed to leptin. In that study, the hypertrophic effects of leptin (1 to 1000 ng/mL) administered for four hours to neonatal rat ventricular myocytes were attenuated by the endothelin receptor A (ET_A_) antagonist ABT-627 as well as the antioxidant catalase [42]. As the ET_A_ antagonist attenuated both the hypertrophic response as well as the increased ROS production following leptin addition, the authors proposed that the hypertrophic response to leptin was dependent on ET_A_ receptor activation secondary to ET-1 upregulation resulting in increased ROS production producing the subsequent hypertrophic response [42]. 

### 6.3. RhoA/ROCK Pathway

The RhoA/ROCK pathway is important in the regulation of a large number of cellular functions related to both physiology and pathophysiology of numerous organs including constituents of the cardiovascular system. RhoA is a key member of the Rho GTPase family, which in turn activates its downstream effector ROCK (either ROCK1 or ROCK2) belonging to the family of serine/threonine kinases, which can phosphorylate a large number of substrates that may contribute to the myocardial remodelling process. Thus, the RhoA/ROCK system plays an important role in blood pressure regulation as well as having a direct influence on various forms of cardiac pathologies including myocardial remodelling, which can contribute to heart failure [62,63]. As recently reviewed, pharmacological inhibition of ROCK or genetic deletion attenuates the degree of myocardial remodelling in various experimental models, thus adding credence to the concept of RhoA/ROCK activity as a contributor to the cardiac hypertrophic and remodelling processes [63].

Our laboratory has used a number of experimental approaches to clearly demonstrate a key role of the RhoA/ROCK pathway in mediating the hypertrophic effects of leptin. For example, we have shown using neonatal rat ventricular myocytes that leptin produced a marked activation of RhoA in these cells, which was blocked by an OBR antibody [64]. Moreover, the hypertrophic effect was similarly blocked by the RhoA and ROCK inhibitors C3 exoenzyme and Y-27632, respectively [64]. We attributed the ROCK-dependent hypertrophic effect of leptin to increased polymerization of actin, as reflected by a decrease in the G/F-actin ratio, which occurs as a result of LIM kinase-dependent cofilin phosphorylation, as illustrated in Figure 5 [64]. Our studies also suggested that intact caveolae, which are a subset of lipid rafts and characterized by flask-shaped invaginations and are rich in sphingolipids, cholesterol, and various caveolin proteins, are critical for RhoA/ROCK-dependent leptin-induced cardiomyocyte hypertrophy [65]. Leptin profoundly increased caveolae expression in cardiomyocytes as determined using molecular analyses and electron microscopy [65]. Importantly, it appears that OBR is co-localized with caveolae whereas caveolae disruption with the cholesterol-chelating agent methyl-β-cyclodextrin (MβCD) completely prevented the pro-hypertrophic effect of leptin [65]. Furthermore, and of relevance to the previous section, the pro-hypertrophic effects of leptin in NRVMs were associated with a selective translocation of p38 MAPK into nuclei, which was RhoA- and caveolae-dependent as evidenced by significant attenuation of p38 MAPK nuclear translocation by MβCD as well as pharmacological inhibition of RhoA and ROCK [65]. It is also important to note that MAPK involvement in leptin-induced hypertrophy was found to be restricted to p38-dependent effects. Thus, although the ERK1/2 import into nuclei was also increased following leptin addition, this was unaffected by either caveolae disruption or RhoA or ROCK inhibition, suggesting a dissociation between leptin-induced hypertrophy and ERK1/2 activation [65]. A similar observation was observed in a study showing that the vascular hypertrophic effect of leptin was inhibited only by a p38 inhibitor but not by other MAPK inhibitors, reinforcing the concept of p38 MAPK involvement in the hypertrophic response to leptin [66].

Although the precise downstream mechanism for RhoA/ROCK-dependent leptin-induced cardiomyocyte hypertrophy is not fully understood, it appears that this may occur via a multiplicity of mechanisms. Among these is activation of the kinase mammalian target of rapamycin or mTOR and the phosphorylation of its major target p70(S6K), which in turn would result in the activation of GATA4, a major transcriptional factor involved in the hypertrophic program [46]. Related to this finding of GATA4 activation is the observation of a RhoA/ROCK-dependent stimulation of the calcineurin pro-hypertrophic factor following leptin administration. Calcineurin is a serine/threonine protein phosphatase, which plays a key role in the hypertrophic program. It is activated by increased intracellular calcium levels and the formation of a calcium–calmodulin complex, which results in the dephosphorylation of the transcriptional factor Nuclear Factor of Activated T cells (NFAT), resulting in NFAT, and more specifically the NFAT3 isoform, translocation to the nucleus where it interacts with various transcriptional factors including GATA4 to promote the hypertrophic response [67]. We have reported that leptin-induced Rhoa/ROCK activation results in calcineurin activation and the resultant NFAT translocation into nuclei in cultured neonatal rat ventricular myocytes [68]. Surprisingly, the RhoA/ROCK-dependent calcineurin activation occurred via both calcium-dependent and calcium-independent mechanisms.

It is interesting to also note that ginseng, a Traditional Chinese Medication demonstrating excellent antihypertrophic effects as well as an ability to attenuate heart failure in animal models [69], can effectively inhibit leptin-induced hypertrophy in neonatal cultured rat ventricular myocytes [45]. This appears to reflect the ability of ginseng to prevent the leptin-induced activation of Rho guanine nucleotide exchange factor 1 (p115RhoGEF), a GTPase activating protein, resulting in reduced ROCK activation and p38 MAPK translocation into nuclei [45]. A summary of leptin-induced RhoA/ROCK-dependent processes potentially contributing to the pro-hypertrophic effect of leptin is illustrated in Figure 5.

It is also of relevance to point out as a corollary that leptin has been shown to produce cytoskeleton remodelling in nucleus pulposus cells as well as in chondrocytes through the RhoA/ROCK pathway [70,71]. Although unrelated directly to the cardiovascular system, these findings may be of importance in understanding the mechanisms underlying obesity-associated lumbar disc degeneration as well as the development of osteoarthritis [70,71]. Moreover, it suggests that leptin-induced RhoA/ROCK activation likely represents a multiorgan and multitissue phenomenon. 

### 6.4. Upregulation of Cardiomyocyte Leptin Production by ET-1 and Angiotensin II May Mediate the Pro-Hypertrophic Effects of Both Factors through Activation of NF-κB and p38 MAPK

As already alluded to, both angiotensin II and ET-1 can significantly increase cardiac leptin production. This increase in leptin production likely contributes, at least in part, to the hypertrophic effects of both angiotensin II and ET-1 in an autocrine manner and paracrine manner. The ultimate hypertrophic response is likely dependent on the activation of the transcriptional factor kappa-light-chain-enhancer of activated B cells (NF-κB), which in turn activates p38 MAPK. This concept is based on various lines of evidence. First, the hypertrophic effects of both angiotensin II and endothelin-1 were associated with increased leptin secretion and gene expression in neonatal rat ventricular myocytes concomitant with significantly increased NF-κB phosphorylation, increased translocation of NF-κB into nuclei, as well as increased NF-κB-DNA binding activity following addition of angiotensin II or endothelin-1 [72]. Secondly, inhibition of NF-κB significantly blunted both the angiotensin II- and endothelin-1-induced p38 MAPK activation whereas, thirdly, inhibition of p38 MAPK blocked both angiotensin II- and endothelin-1-induced elevation in leptin production.

### 6.5. Leptin and FTO

Another possible mechanism by which leptin could stimulate the hypertrophy program is through the activation of the fat mass and obesity-associated (FTO) gene, which has been shown to be closely related to obesity in experimental animals. Specifically, upregulation of the FTO protein, which functions as an N6-methyladenosine (m6A) demethylase, in mice is associated with body weight gain whereas its deletion produces weight loss [73,74]. While the precise mechanisms underlying the ability of FTO to modulate body weight are not known, a substantial contribution likely reflects the ability of FTO to regulate energy expenditure via the hypothalamus [75]. 

Although the initial primary role identified for FTO is related to energy expenditure and adiposity, recent evidence suggests that FTO is likely involved in various cardiovascular pathologies most likely secondary to changes in m6A methylation, the primary target of FTO in the nucleus and an important contributor to cardiac biology [76,77]. With respect to myocardial remodelling, current evidence obtained from both animal and clinical studies suggests that FTO is downregulated in the failing heart (as well as hypoxic cardiomyocytes) whereas upregulating FTO levels improves contractile function [78,79]. On the other hand, inhibition of FTO has been shown to reduce cardiomyopathy associated with a four-week high-fat-diet-induced hyperlipidemia in rats [80]. Thus, FTO appears to function as an endogenous cardioprotective or cardiodeleterious factor during myocardial hypertrophy and remodelling, which may be dependent on the mode of insult and potentially other factors. While there is a paucity of data documenting the potential role of FTO in leptin-induced cardiomyocyte hypertrophy, work from our laboratory suggests that FTO may function as a contributing factor to leptin-induced hypertrophy, at least in cultured ventricular myocytes. In these studies, leptin addition to neonatal rat ventricular myocytes resulted in a cardiomyocyte hypertrophic response, which was associated with FTO upregulation in terms of both gene and protein expression [48]. Importantly, the hypertrophic response to leptin was abrogated when FTO was knocked down using small interfering RNA [48]. Also interesting was the finding that FTO upregulation was seen only with leptin as the pro-hypertrophic factor with no effect on FTO seen when endothelin-1 or angiotensin II were used as the pro-hypertrophic factors, suggesting that any contribution of FTO in the hypertrophic process is selective for leptin as the hypertrophic stimulus. It remains to be determined whether FTO regulates in any way the hypertrophic and remodelling processes using in vivo models of heart failure. 

### 6.6. Potential Role of the JAK/STAT Pathway

As noted earlier in this review, the JAK/STAT pathway contributes to biological responses in numerous cell types and is a primary cell signalling response following cytokine receptor activation [16]. This is illustrated in Figure 2 and demonstrates the ability of leptin in particular to modify gene expression, which would be of relevance to the development of cardiac hypertrophy. With respect to this, it is unclear as to whether JAK/STAT activity contributes to or limits hypertrophic responses to stimuli. Indeed, it has been shown that the hypertrophic response to doxorubicin is enhanced in STAT3-overexpressing mice whereas cardiotoxicity is attenuated with improved survival rates in these animals [81]. A number of studies have shown that inhibition of cardiac hypertrophy and remodelling in different experimental models is associated with inhibition of JAK2/STAT3 activity [82,83,84]. Conversely, cardiac-specific genetic deletion of JAK2 in mice produces severe hypertrophy and dilated cardiomyopathy, which were associated with left ventricular dysfunction [85], a finding in agreement with an earlier study showing a remodelling and heart failure phenotype in mice with cardiac-specific STAT3 knockout [86]. Moreover, cardiac-specific deletion of the myocardial suppressor of cytokine signaling-3 (SOCS-3), the endogenous feedback inhibitor of STAT3, results in reduced myocardial remodelling and severity of heart failure in the 14-day post-infarcted mouse heart [87].

Based on the preceding discussion, it is apparent that the precise role of the JAK/STAT pathway in the myocardial remodelling and hypertrophic process deserves further studies but it appears the nature of its contribution may reflect the nature of the hypertrophic stimulus. With respect to leptin, however, there is a paucity of data linking leptin-induced cardiac hypertrophy to JAK/STAT activation. In an earlier study, the ability to induce the hypertrophic response in cultured neonatal rat ventricular myocytes exposed to a 48 h leptin administration was associated with STAT3 activation as demonstrated by increased STAT3 phosphorylation and its increased nuclear translocation [85]. Moreover, the hypertrophic response to leptin in these cells was abrogated by the JAK2 inhibitor AG-490 [88]. A dissociation between ObR activation and cardiac hypertrophy as a result of high-fat-diet-induced obesity in mice or in *db*/*db* obese mice has also previously been shown. In that report, hypertrophy was evident in obese mice including those with ObR mutations although STAT3 activation was reduced [89]. The ability to demonstrate hypertrophy in mice with inactivated ObR despite reduced ObR-dependent signalling suggested other underlying mechanisms of hypertrophy unrelated to leptin or STAT3, at least in these obesity models [89].

## 7. Autophagy as a Target for Leptin-Induced Cardiac Hypertrophy

An area of research that has received substantial attention in the past few years is the role of autophagy, also referred to as self-digestion, as a mediator of cardiac hypertrophy and remodelling. The catabolic function of autophagy can be either physiological or adaptive as well as pathological or maladaptive, the latter being activated by cardiotoxic pro-hypertrophic factors such as oxidative stressors and cytokines. The regulation of autophagy is exceedingly complex but the system is under substantial regulation by a host of intracellular signalling factors [90,91]. By removing unnecessary cellular degradation products, autophagy is considered essential for cell and organism survival and has been shown to protect mitochondria during the development of heart failure [92]. However, maladaptive autophagy has been shown to be associated with various forms of cardiac pathologies including the development of myocardial remodelling and heart failure and therefore modulation of autophagy has been proposed as a potential approach towards mitigating cardiac hypertrophy [93,94]. How autophagy or inhibition of autophagy contributes to the hypertrophic program remains poorly understood and indeed, somewhat controversial. While it is beyond the scope of this review to discuss this issue in detail, various excellent reviews can be recommended for the interested reader [93,94,95]. Briefly, inhibition of autophagy can be initiated by pro-hypertrophic factors such as angiotensin II [96] although angiotensin II has also been shown to stimulate autophagy in neonatal rat ventricular myocytes, which was associated with reduced apoptosis in angiotensin II-treated cells [97]. Autophagy inducers such as rapamycin have been shown to reduce myocardial hypertrophy [98]. Thus, autophagy appears to play a dual role in the regulation of the hypertrophy program, which is dictated by numerous factors such as the nature/severity or the type of stimuli of the hypertrophic or remodelling response.

Is there evidence for leptin involvement in the autophagic process? Generally speaking, adipokines have been proposed as regulators of autophagy in both a stimulatory and inhibitory manner [99]. There is emerging evidence that leptin can induce autophagy in cultured HeLa as well as in various tissues including the heart following parenteral administration in mice [100]. However, the relevance to cardiac pathology, particularly as pertaining to myocardial remodelling, is uncertain since evidence derived from studies using cardiovascular tissues is less robust with only a few reports of leptin-induced regulation of the autophagic pathway particularly as this pertains to cardiac pathology and myocardial remodelling. Interestingly, however, in vivo infusion of leptin in mice resulted in the stimulation of autophagy in the heart of these animals as well as in other tissues when compared to basal values [100]. In mice subjected to heart failure produced by thoracic aortic banding, deletion of the endothelial leptin receptor produced an improvement in cardiac function as assessed using echocardiography after eight or twenty weeks of aortic banding [101]. These findings were associated with reduced cardiac fibrosis and enhanced angiogenesis, which were related to decreased production of pro-hypertrophic cell signalling and increased endothelial autophagy in ObR KO animals. This study is of particular interest as it demonstrates an important cross talk between vascular endothelial cells and cardiomyocyte hypertrophy mediated by leptin signalling and further shows that leptin signalling in endothelial cells contributes to cardiac dysfunction after a chronic pressure overload by reducing cardiac angiogenesis and contributing to various aspects of maladaptive hypertrophy, which are reduced by endothelial ObR deletion [101]. 

In addition to the study just discussed, leptin-induced dysfunction in cardiomyocyte shortening was reported to be dependent on the presence of autophagy as evidenced by inhibition of the leptin response by the autophagy inhibitor 3-methyladenine [102]. Moreover, leptin was found to promote autophagy as evidenced by enhanced levels of the autophagosomal marker LC3-II and the pro-autophagic factors Beclin and Atg 5 [102]. In contrast, the effects of leptin in reducing shortening and calcium regulation of cultured adult rat ventricular myocytes were found to be associated with an attenuation of autophagy as assessed using decreased LC3-II and Beclin-1 levels [103]. All of the abnormalities were significantly attenuated by the antioxidant apocynin, tempol, or rapamycin.

## 8. Mitochondrial Function and Dynamics in the Development of Cardiac Hypertrophy

A related process to autophagy is mitophagy, which represents the removal of damaged mitochondria, thus contributing to the maintenance of cellular homeostasis [104]. As for autophagy, mitochondrial mitophagy also represents a complex phenomenon, which is dependent on various cell signalling pathways. Among the most important is mitochondrial PTEN-induced kinase 1 (PINK1), a serine/threonine-protein kinase which causes mitophagy via the Parkin protein, which binds to damaged mitochondria, resulting in the mitophagic process [105]. There is some evidence from studies using cultured cancer cells that leptin can regulate the mitophagy process [106]. It has been proposed that the intracellular ObRB domain binds to mitochondria and thus limits the mitophagy in a variety of cultured cells [107]. However, there is no clear evidence that leptin can regulate mitophagy in the cardiac cell, either in an inhibitory or stimulatory manner. However, as will be discussed below, leptin can directly modulate cardiac mitochondrial function and structure, including enhancing mitochondrial fission, which can contribute to the stimulation of mitophagy as well as the hypertrophic program.

Although mitochondrial injury and dysfunction are well-established consequences of cardiac pathology including heart failure, increased evidence points to the contribution of damaged mitochondria as an important contributor to the cardiac hypertrophic response and myocardial remodelling [108,109,110]. While mitochondrial contribution to the hypertrophic process may involve multiple mechanisms, here, we focus primarily on mitochondrial dynamics, which may be particularly relevant as a target for leptin-induced modulation. As recently reviewed, two particularly important aspects of mitochondrial dynamics involve mitochondrial fusion and fission, which contribute to various aspects of cardiac pathology [111]. As illustrated in Figure 6, mitochondrial fission and fusion represent dynamic processes by which two mitochondria merge into one or one mitochondrion separates into two, respectively [112]. Fusion is coordinated primarily by the proteins mitofusin (MFN) 1 and 2 located on the outer mitochondrial membrane and optic atrophy (OPA) 1 located on the inner mitochondrial membrane whereas mitochondrial fission is mediated primarily by dynamin-related protein1 (Drp1), a pro-fission protein that opposes Mfn2 and is also of importance in the mitophagic process. Prevention of mitochondrial fusion or fission has been shown to enhance cardiac hypertrophy and overall pathology, strongly suggesting that an imbalance between mitochondrial fission and fusion leads to a cardiac pathological response [113,114,115,116].

In theory, leptin may participate in the modulation of mitochondrial function and structure via two processes, first by stimulating cell signalling, which targets mitochondria, and secondly by directly targeting mitochondria, which can occur through intracellularly derived leptin. The ability of leptin to target mitochondria has been known for some time particularly as this relates to adipose tissue where leptin has been shown to target uncoupling proteins, resulting in an enhanced proton leak and altered energy metabolism [117]. The identification of functional leptin receptors in cardiac mitochondria adds credence to implicating mitochondria as direct targets for intracellularly produced leptin [118]. Interestingly, however, leptin had no effect on mitochondrial structure or function under normal conditions but significantly enhanced the ability of calcium to produce mitochondrial swelling in cultured neonatal rat ventricular myocytes, an effect attenuated by a leptin receptor antagonist [118]. Leptin has been shown to stimulate apoptosis in cultured neonatal rat ventricular myocytes secondary to a calcium-dependent increase in mitochondrial permeability transition pore opening stimulated by leptin in these cells [47], thus potentially contributing to apoptosis-dependent cardiac remodelling via this process. Various parameters of mitochondrial dysfunction have also been reported in studies using isolated perfused rat hearts treated with leptin for up to four hours as manifested via uncoupling of oxidative phosphorylation and loss of membrane potential [119].

## 9. Conclusions

Since its initial discovery in 1994, research interest in leptin has increased dramatically not only with respect to its function as a satiety factor but also in relation to its ability to modulate physiology and pathophysiology of various organ systems. This is clearly evident with respect to the heart, which has been shown to be a source of leptin production and to function as a target for leptin’s effects. Substantial evidence has been presented in the literature that leptin functions as a cardiac hypertrophic factor and thus can contribute to heart disease, particularly heart failure, under conditions of hyperleptinemia such as that seen in obesity. As discussed in this review, it appears that the pro-hypertrophic effect of leptin is likely mediated by multifaceted mechanisms involving molecular and intracellular changes as evidenced by modulation of cell signalling pathways as well as such cellular processes as autophagy and mitochondrial dynamics. Substantial future work is necessary to more precisely delineate the molecular and cellular mechanism(s) underlying the hypertrophic effects of leptin using physiologically relevant experimental models. Based on expanding knowledge of ObR structure, regulation, and activation, this work could be assisted by the continuing development of highly specific ObR antagonists used to probe leptin-induced cardiac effects as well as enhancing the possibility of their future development as pharmacological agents.

## Figures and Tables

**Figure 1 ijms-25-01137-f001:**
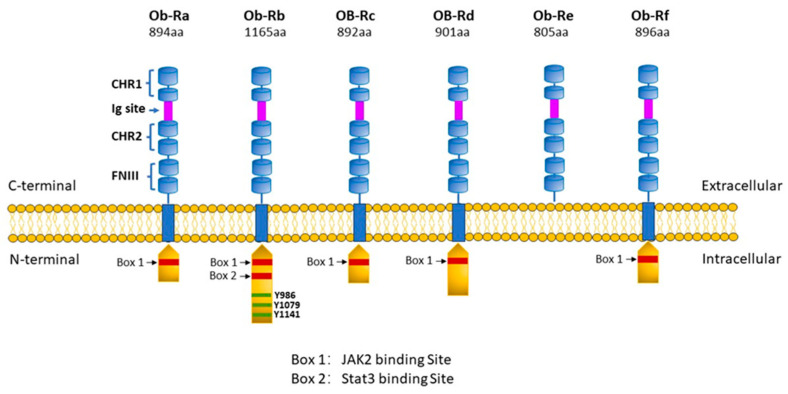
Leptin receptor subtypes. Six different isoforms of ObR have been identified, denoted as ObRa, ObRb, ObRc, ObRd, ObRe, and ObRf. Each receptor subtype shares two cytokine-binding homology regions (CHR1 and CHR2) with CHR2 representing the main binding site for leptin, an IgG-like domain, and two fibronectin type 3 domains (FN3) within its C-terminal. All isoforms have transmembrane regions except the ObRe receptor, which functions as a soluble leptin binding receptor not anchored to the cell membrane and which binds circulating leptin, thereby regulating leptin bioavailability and functionality but is unable to transduce any downstream signalling. The intracellular N-terminal domain expresses a BOX-1 motif, which is critical for JAK-2 binding. The OBRb receptor is the only long isoform to also express a BOX-2 motif, which facilitates the activation of the JAK2/STAT transduction pathway. Moreover, ObRb contains 3 tyrosine residues (Tyr986, Tyr1076, and Tyr1141) whose phosphorylation enables STAT3/STAT5 activation. The short isoforms ObRa,c,d,f have only one intracellular domain (Box 1) and therefore limited signalling capacity. Created with PowerPoint software (Microsoft Office 365).

**Figure 2 ijms-25-01137-f002:**
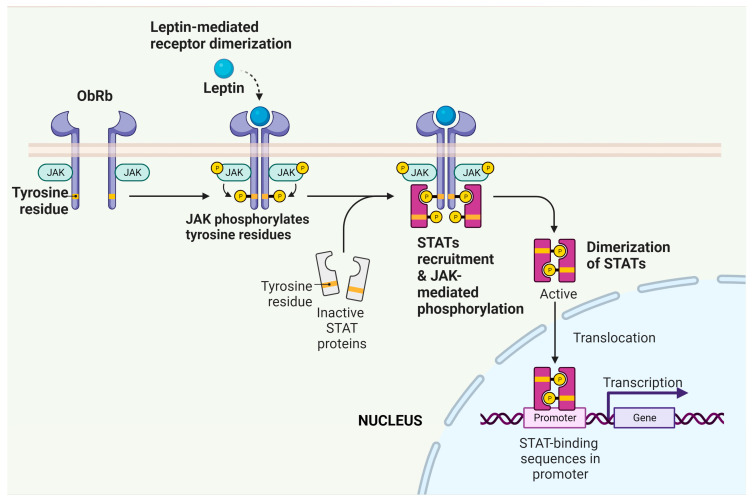
Activation of the JAK/STAT pathway by leptin. Binding of leptin to ObRb on target cells results in receptor dimerization, which results in the ability of associated JAKs to phosphorylate one another. These trans-phosphorylated JAKs can now phosphorylate a number of targets including STATs, which can now enter the nucleus and regulate the transcription of target genes. Created with BioRender.com.

**Figure 3 ijms-25-01137-f003:**
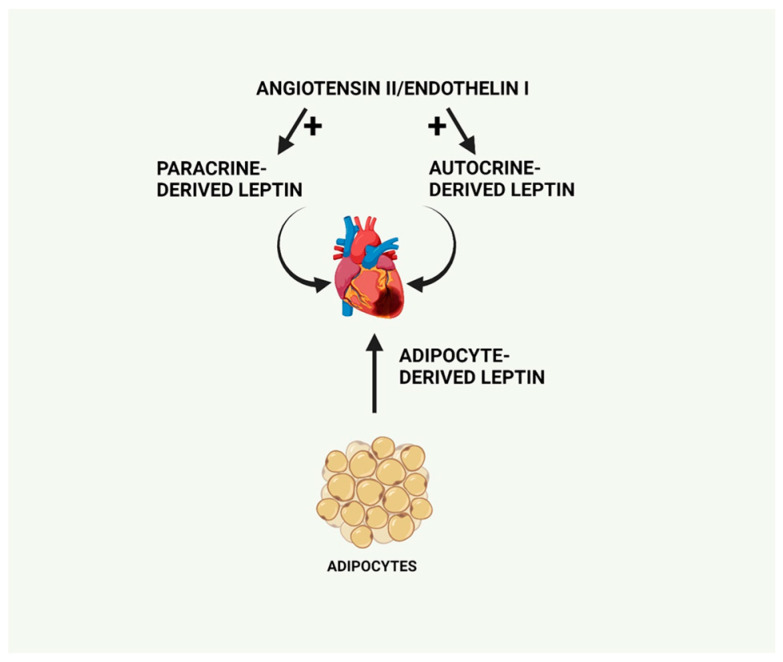
Primary sources of leptin production, which can exert cardiac effects including circulating adipocyte-derived leptin as well as locally produced leptin, functioning in a paracrine or autocrine manner and stimulated by pro-hypertrophic factors including angiotensin II or endothelin-1. Created with BioRender.com.

**Figure 4 ijms-25-01137-f004:**
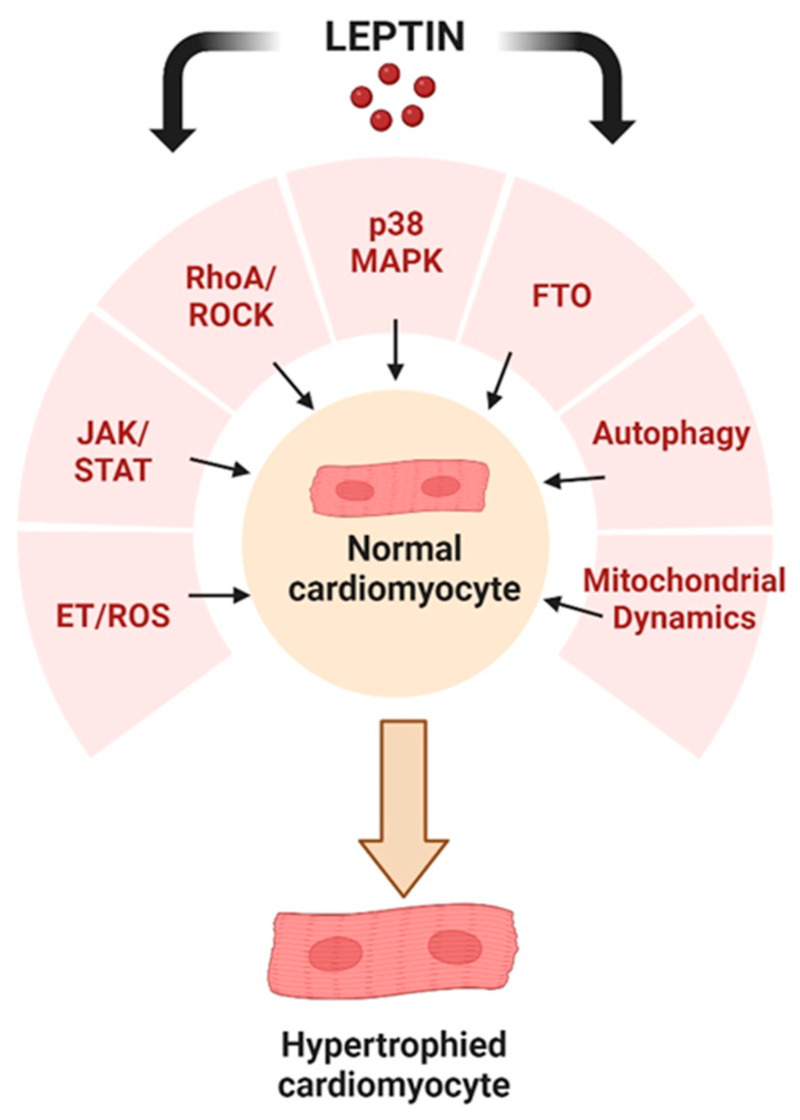
Potential cellular events underlying the pro-hypertrophic effects of leptin as discussed in this review. Created with BioRender.com.

**Figure 5 ijms-25-01137-f005:**
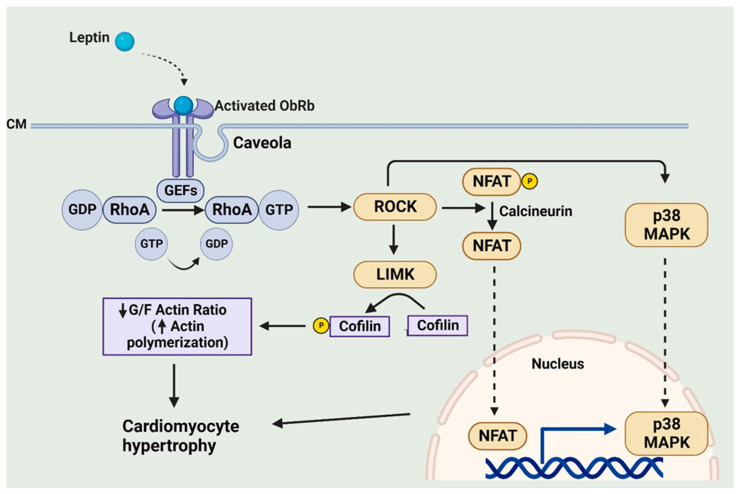
Leptin-induced caveolae-dependent activation of RhoA/ROCK and its contribution to the hypertrophic program. CM, cell membrane; GEFs, guanine nucleotide exchange factors; LIMK, LIM kinase. Created with BioRender.com.

**Figure 6 ijms-25-01137-f006:**
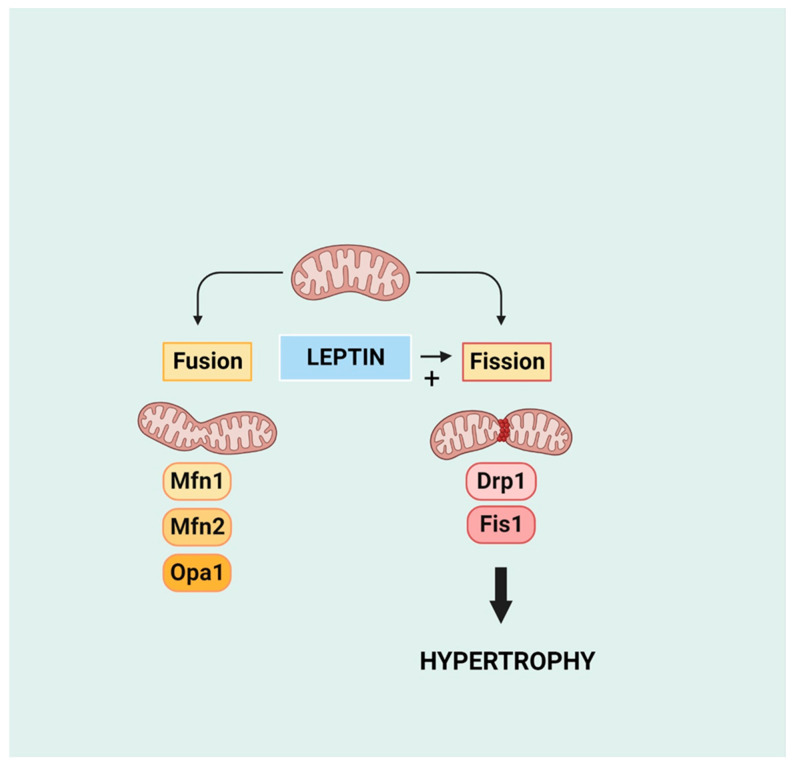
Mitochondrial dynamics in terms of fusion and fission and the pro-fission effect of leptin. Mfn 1 and 2 and Opa1 are pro-fusion proteins mitofusin 1 and 2 and optic atrophy 1 protein, respectively. Drp1 and Fis1 are pro-fission proteins dynamin-related protein 1 and mitochondrial fission protein 1, respectively. Created with BioRender.com.

**Table 1 ijms-25-01137-t001:** List of various major and minor adipokines secreted by adipose and other tissues.

Adipokine	Selected Sites of Synthesis
Leptin	Adipocytes, cardiomyocyte, vascular endothelial cells
Adiponectin	Adipocytes, skeletal muscle, cardiomyocytes
Apelin	Adipocytes, cardiomyocyte, pancreas, stomach
Chemerin	Adipocytes, liver, lung, placenta
Resistin	Adipocytes, blood monocytes, liver
Visfatin	Adipocytes, lymphocytes, macrophages
Omentin	Adipocytes (primarily visceral)
Vaspin	Adipocytes, hypothalamus, stomach, liver, pancreas
Progranulin	Adipocytes, CNS
Interleukin-6	Adipocytes, t cells, macrophages, fibroblasts
MCP-1	Adipocytes, macrophages, epithelial cells, endothelial cells
PAI-1	Adipocytes, vascular endothelium, macrophages, cardiomyocyte
RBP-4	Adipocytes, macrophages, liver
TNFα	Adipocytes, macrophages, endothelial cells, cardiomyocytes
CTRP-4	Adipocytes, brain, skeletal muscle

Table summarizes adipokines thus far identified and summarizes selected sites of synthesis although numerous additional sources for individual adipokines have been reported. CNS is the central nervous system; MCP-1 is monocyte chemotactic protein-1; PAI-1 is plasminogen activator inhibitor-1; RBP-4 is retinol binding protein-4; TNFα is tumor necrosis factor-alpha; CTRP-4 is C1q/TNF-related protein-4.

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
