# Peer review of "Molecular and Cellular Mechanisms Underlying the Cardiac Hypertrophic and Pro-Remodelling Effects of Leptin"

_ijms, 2024, doi:10.3390/ijms25021137_

Round 1

Reviewer 1 Report

Comments and Suggestions for Authors

This a very-well written and comprehensive review on the role of leptin in cardiac hypertrophy and discusses the various mechanisms implicated in this effect. The information provided in this paper are very important and as it summarizes the literature in the field and much more. The range of references used is adequate and up to date. The figures are clear, and the legends are well-detailed. Finally, the corresponding author is a prominent and internationally recognized scientist in the field of cardiac hypertrophy and heart failure.

The reviewer has only very minor comments about this review:

1-      Table 1: Title, please provide more details on «other tissues». If possible, it would also be very informative, if the authors list where the various adipokines are secreted from.

2-      Figure 1: At the end of the legend, please add created with BioRender.com and if it is not so, where is comes from.     

Author Response

We wish to thank the reviewer for his/her positive comments.  We have revised the manuscript as requested. Please note that all changes are highlighted in yellow.

  1. We have modified Table 1 and provided key sources of adipokine secretion in addition to adipocytes. In essence, there are numerous additional sources of adipokine production secretion and we have included what may represent the most relevant of the non-adipocyte sources.
  2. Regarding Figure 1, Biorender was not used to create this figure but rather PowerPoint software. This has been added to the figure legend. 

Reviewer 2 Report

Comments and Suggestions for Authors

Comments to the author:
The Manuscript "
Molecular and Cellular Mechanisms Underlying the Cardiac

Hypertrophic and Pro-Remodelling Effects of Leptin. The manuscript discuss mechanisms which potentially mediate leptin induced cardiac pathological hypertrophy which may con tribute to the development of heart failure. Manuscript is well written and presented nicely
Kindly incorporate these points into the manuscript
Comments:
1- Does leptin directly cause cardiac hypertrophy?

2- What is role of leptin in the regulation of cardiac metabolism?

3-Role of leptin in protection against cardiac lipotoxicity?

Author Response

We wish to thank reviewer #2 for his/her positive and constructive comments and have revised the manuscript according as summarized below. Please note that all revisions are highlighted in yellow. 

  1. Indeed, leptin directly induces hypertrophy as demonstrated in studies using cultured myocytes in particular.  This is discussed in the manuscript (lines 231-238) although we have added an additional statement reinforcing this fact.
  2. The question of the contribution of leptin to the regulation of cardiac energy metabolism is of course an important one but the topic is beyond the scope of this revue.  We have addressed this to some degree (lines 200-221) and have added a concluding view regarding the potential regulation of cardiac energy metabolism by leptin ((lines 226-229).  
  3. Related to the preceding comments we have added a brief discussion concerning the potential effect of leptin on cardiac lipotoxicity (lines 218-226)